# A Simulation System Towards Solving Societal-Scale Manipulation

## Abstract

The rise of AI-driven manipulation poses significant risks to societal trust and democratic processes. Yet, studying these effects in real-world settings at scale is ethically and logistically impractical, highlighting a need for simulation tools that can model these dynamics in controlled settings to enable experimentation with possible defenses. We present a simulation environment designed to address this. We elaborate upon the Concordia framework that simulates offline, 'real life' activity by adding online interactions to the simulation through social media with the integration of a Mastodon server. Through a variety of means we then improve simulation efficiency and information flow, and add a set of measurement tools, particularly longitudinal surveys of the agents' political positions. We demonstrate the simulator with a tailored example of how partisan manipulation of agents can affect election results.

## 1 Introduction

Large Language Models (LLMs) are becoming increasingly persuasive [1, 2]. While this can be a positive indicator that they are delivering high quality and compelling responses, it also means they have the ability to manipulate. Even before ChatGPT, sophisticated technology-enabled manipulation was creating large-scale risks and harm [3–5]. Now, with persuasion capabilities surpassing average human levels in many settings [1, 6, 2, 7, 8], there is a worsening threat of severe harm through societal-scale manipulation [9, 10]. Robust ways to mitigate such risks are urgently needed.

However, doing so remains difficult given our inability to effectively and consistently perform experiments, be it simulating attacks and threats or evaluating the effectiveness of defenses against AI-powered manipulation. This lack of experimental control is pervasive in the social sciences, where societal-scale treatments are challenging to implement—e.g., amidst the many confounding factors in misinformation spread and the ethical concerns regarding manipulative human experimentation.

LLMs are the first models capable of replicating, even if at low fidelity, the complexity of human agent behavior. In pursuit of a simulator to test defenses to AI manipulation, we thus leverage these systems, and in particular recent breakthroughs in LLM-based multi-agent simulations [11, 12]. In particular, we take advantage of the Concordia [12] framework, which simulates social systems over the 'in real life' time of agents. However, significant manipulation of the beliefs and behaviors in our society occurs online. This motivated us to combine a revised version of Concordia with a Mastodon server, creating realistic community interactions on a social media platform.

Our contributions include:

- **Mastodon:** We implement an actual social media environment (Mastodon) within Concordia that seamlessly integrates into the everyday experience of the agents.

Submitted to 38th Conference on Neural Information Processing Systems (NeurIPS 2024). Do not distribute.

- **Efficiency & Realism:** We provide an efficient system to simulate social-media activity without unnecessary simulation of offline behavior. We make our implementation efficient and scalable through a combination of cloud infrastructure, selective use of elaboration of context within Concordia, and the parallelization that our design enables. We also create a data pipeline from survey response to sample trait scores when generating agents.
- **Measurement tools:** We provide a custom analytics dashboard of Mastodon social network activity, as well as a longitudinal survey system for the agent population.
- **Demonstration:** Using our system, we create an election simulation example in which we ground agents in demographic-specific scores of social values. We test a control and two alternative versions, each analyzed using our diagnostic tools. We show results of longitudinal surveys addressing political polarization and misinformation.
- **Code:** Our code is available at `https://anonymous.4open.science/r/anon-sim-3EEB`.

## 2  Related Work

There have been many attempts to replicate complex systems in hopes of creating believable, but tractable, simulations that can mimic social dynamics with high fidelity in order to explain or probe specific phenomena. One area of focus has been online environments, such as social media, where issues like misinformation and polarization abound; many efforts [13–19] attempt to simulate these settings with relatively simple social characteristics (e.g., homophily), but without necessarily emulating the full complexity of these online settings. Several works [20–25] also focus on the algorithmic aspects of these online settings by modeling algorithms to directly modulate the likelihood of agent interactions or connections, information visibility, and more.

More recently, given their ability to emulate human-like behavior and responses, a number of works have highlighted the promise of LLM agents for simulations. LLMs have been studied as individuals through diverse lenses such as politics [26], psychology [27, 28], marketing [29], and behavior [30, 31]. While not without limitations [32], these works show the ability of LLMs to reflect realistic individual behavior in many settings. Several works have built frameworks with multiple interacting LLM agents, aiming to produce interesting or realistic phenomena. [11, 33, 12, 34] provide general environments where LLM agents adopt personas and interact amongst themselves in settings such as fictitious towns. Some works [35–38] build social media simulations, providing insights into topics like social movements and news feed algorithms. However, to our knowledge, our work is the first to integrate a real, rather than facsimile, social media platform with the aim of constructing a scalable testbed for solutions to large-scale manipulation. We also are not aware of the use of social values as generative agent traits, though they have been used as feature components for LLM alignment [39].

## 3  Methodology

Our work builds upon the Concordia framework, "a software library developed to simulate interactions of generative-model agents in a grounded physical, social, or digital space" [12]. It relies heavily on LLM-based elaboration of situational and social context using structured text and summarization. Agent models in Concordia are rich and versatile: they have memories, long-term objectives, and even homeostatic drives, all of which are used as rich social context for LLMs to infer agent plans (*e.g.* What kind of person is X? What situation is X in now? What would a person like X do here?). Social system simulation is then performed by a centralized controller (the 'Gamemaster') that evaluates attempted actions, handles events, and distributes their effects. The simulation runs in discrete steps of fixed simulated time intervals.

Specifically, we integrate Mastodon as a phone application for our agents to use within Concordia, adding a realistic form of online interaction and communication among agents. We select, add to, and overhaul many of the Concordia systems to do this in an efficient and scalable way. We then run several election simulations using our framework where we repeatedly poll agents with survey questions.

**Mastodon Integration.** Mastodon is a popular open-source social media platform and offers several ways for users to interact with both content and one another (e.g., posting, following, liking). We built a Mastodon smartphone application within the Concordia environment that provides the user with the option to use the app. Separately, we set up Mastodon on a cloud server with a number of blank-slate users. With this setup, our simulation begins by first building the agents within Concordia and then assigning the blank-slate Mastodon users to them, modifying user profile information respectively, and then sampling an initial followership network. Agents then take full control of their accounts,

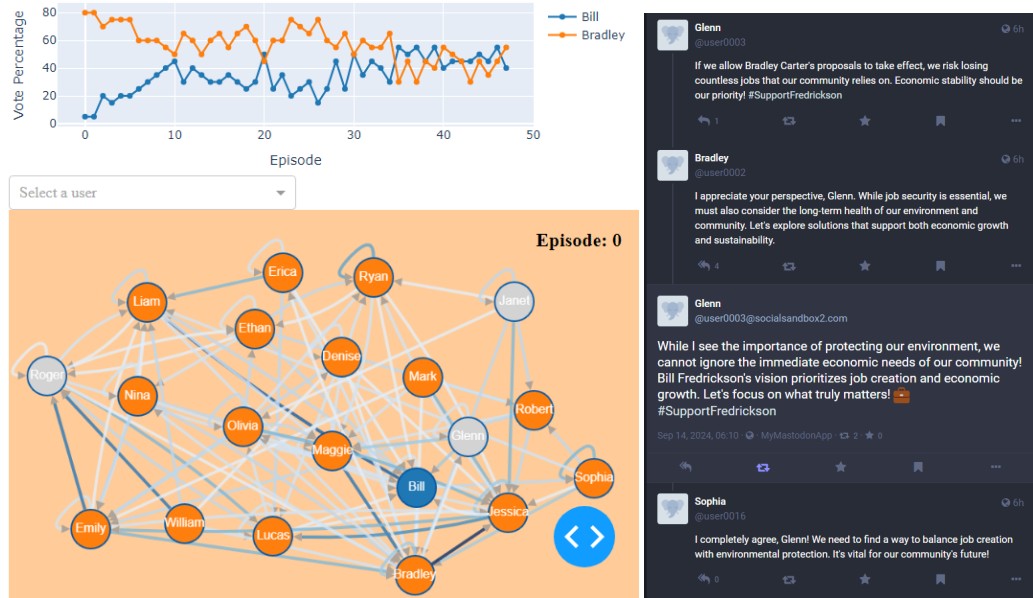

Figure 1: **Illustration of simulator.** Left: A snapshot of our Mastodon analysis dashboard showing vote percentage over time on top and the Mastodon social network at an episode (here episode 0) below. Node colors (same as top) show vote preference. Right: A snapshot of the current timeline of one selected agent from the simulation in which we included a malicious agent (Glenn) whose goal is to convince voters to support Bill Fredrickson over Bradley. (data from the $N = 20$ control experiment).

which enables them to modify all aspects in the context of natural behaviour on the platform. By default, our code randomly generates a network where the frequency of symmetric connections (when agents follow each other) is higher than a fully random network to account for homophily observed in real social networks [40] (see appendix C for exact procedure). To populate the platform with initial content to engage with, each agent makes an introductory post[1]. While agents make their own decisions, to control the volume of activity on the platform, we fix a base rate of per-agent app opening that is supplemented by an additional probability of taking an action at each episode. When agents open Mastodon, they read their feed and choose from a set of actions ('post', 'follow', etc.).

**Efficiency & Scaling.** To efficiently scale up social media environment simulations, we also present a system design that simulates only the social media environment, forgoing the real-world aspect. This allows us to replace elements of Concordia responsible for simulating the world and coordinating the world state with ones dedicated to online experience, further allowing us to parallelize several steps of within-agent processing in the simulation. Together with the probabilistic generation of fixed time steps at which every agent acts, this yields significant benefits in both cost and time. Other aspects of efficiency improvements include selectively choosing important components such as those describing an agent's persona, while removing components unnecessary for the social simulation such as their somatic state. Our current implementation runs a simulation of 24-hours ($\Delta t = 30$ min.) for a 20-agent system in under 2.5 hours at the cost of 10 USD using OpenAI's GPT-4o-mini. This marks a 70% decrease from the >8 hour times observed before the improvements. It also scales up to 100 agents with a run-time of around 3 hours.

**Measurements.** Over the course of our simulations, we implemented a longitudinal survey of agents that asks each agent at each time step of the simulation a set of questions. Our survey questions and formats are similar to those typically found in political survey research [41]. For instance, when surveying the agents on candidates' favorability, we ask agents to answer on a scale of 1 to 10, with 1 representing strong dislike and 10 strong favorability. When asking agents for voting preferences, we simply ask the agent to answer with the candidate's name. The exact prompts used can be found in the Appendix A. Our framework also allows additional, custom survey questions to be easily added.

---

[1] A 'toot' in Mastodon terminology

# 4 Analysis

**Agent Personas.** The simulation defines an agent's persona through a given set of quantified personality traits (i.e., a trait set) along with an additional set of natural language statements expressing deep-rooted values or deeply-held beliefs. In social sciences and psychology, trait sets can be operationalized as scores on features derived from responses to well-validated survey questions. There is a conditional distribution of score sets over respondent demographic information such as age and gender. We provide a pipeline to process survey responses into trait scores given the scoring map. The Concordia library's default example use random values for the Big-5 personality traits used in psychology, potentially generating unrealistic distributions of agent behaviour. Social values are a related, but distinct set of latent features predictive of behavior. We add to our system a toggle for replacing Big-5 with the standard set of social values [42] and setting their scores using published demographic-conditioned survey data [43] that uses a well-validated 20-question survey [44].

These score sets and demographic information are passed to Concordia's agent generation procedure, which takes them, along with name, gender, goal, context, and formative experiences, to generate autobiographical anecdotes that are then summarized by an LLM to form a backstory describing the agent's complete life. These are then transformed into a series of formative memories for the agent that are passed to the agent as observations.

**Election Demonstration & Setup.** For our election setting, we configure the generic knowledge possessed by the agents and gamemaster (GM) within Concordia to include information about the fictitious town of Storhampton in which they live, several social/economic issues facing Storhampton, the upcoming mayoral election, and general knowledge about Mastodon (see appendix B for specific texts). This shared context is added to the formative memories of all agents. We introduce several agent types as subjects of study.

- General Voting Agents: We built "Opinion on candidate" components, where the agent retrieves relevant information from its long-term memory and summarizes it to state their general opinion of each candidate. We use these results alongside the agent's most recent observations, and their personas, to create a verdict on the agent's "Current Opinion on candidate" by explicitly prompting the agent to consider recent events and their effects on the voter's perception of a candidate.
- Candidate Agents: Candidate agents consist of components that retrieve relevant memories about the evaluations of both the candidates and their opponents. The results from these are used alongside the candidate's persona to give the agent context to come up with a "Plan to improve Public Perception" for their campaign. Note that we selected two male candidates to control for gender in the experiment.
- Malicious Agents: these are similar to the candidate agents, but they are prompted instead to develop a strategy to harm the opposing candidate and improve the perception of the favored candidate using disinformation and other underhanded means.

The candidates campaign and policy proposals is shared with all other agents and appended to their memory. In what follows, we explore how simulation outcomes vary with the degree of partisan alignment expressed in the candidate's policy proposals.

**Simulations.** To demonstrate our simulation framework, we conduct three simulation runs with $N = 20$ and $N = 100$ agents each over the span of 24 hours of in-simulation time divided into 30 minute episodes using OpenAI's gpt-4o-mini language model. We fix each agent's base social media usage rate to 5 times per day, with times randomly sampled from the 48 possible time steps during which the agent can act. We also add a stochastic usage pattern where agents access the app at a per-step rate of $p = 0.15$ per episode. For all simulations, we generate a fixed set of agent configurations, with two mayoral candidate agents each having partisan policy proposals related to the city. Candidate Bill campaigns on "providing tax breaks to local industry and creating jobs to help grow the economy.", while Bradley campaigns on "increasing regulation to protect the environment and expanding social programs.". We conduct the following simulations:

1. **Simulation 1: the control case.** Here, we provide all agents other than the candidates with a simple goal inspired by an example in the Concordia codebase (i.e., to "have a good day and vote in the election"). They are provided with no other context beyond a set of randomly generated Big-5 persona traits that are unchanged throughout this set of simulations.
2. **Simulation 2: the bias case.** This adds a belief to all non-candidate voters that is biased towards Bill's policy proposals: the agents are initialized with the context that they "don't care about

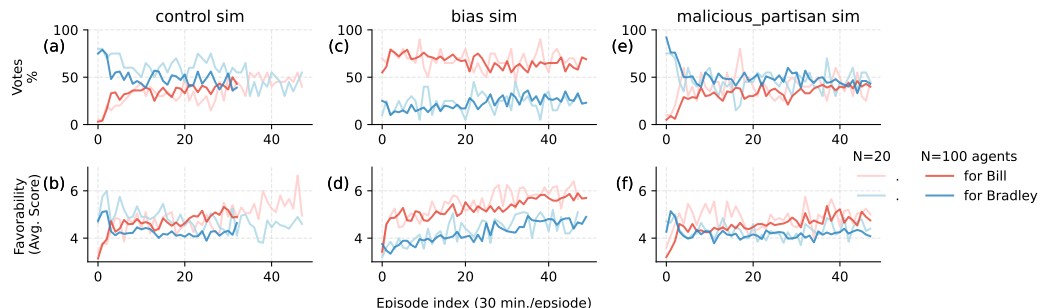

Figure 2: **Longitudinal survey results.** Vote percentage is shown on top and average candidate favorability on bottom for each experiment type: a control setting (panels (a) and (b)), a biased voter setting (panels (c) and (d)), and a malicious partisan setting (panels (e) and (f)), respectively, for Bill (red) and Bradley (blue). Light shades are for a simulation with $N = 20$ agents; dark shade for $N = 100$.

172     the environment, only about having a stable job". Everything else remains the same as in the
173     preceding simulation.

174 3. **Simulation 3: the malicious case.** We alter a single agent (Glenn) to be a malicious partisan
175     for Bill. This agent is initialized with the goal of strongly advocating for Bill, while convincing
176     others to support Bill using manipulation such as spreading disinformation. Additionally, we give
177     this agent a slightly higher base social media usage rate of 10, to better evaluate the impact of the
178     malicious agent.

179 We first illustrate the experimental output generated by our system. In fig. 1, we provide a snapshot
180 of our dashboard used for social media network analysis as well as a snapshot of the Mastodon
181 timeline of a randomly sampled agent. The dashboard shows user interactions, and one can click on
182 individuals to focus on them and investigate their influence. The example Mastodon timeline shows
183 how in one of our manipulation experiment the malicious agent (Glenn) seems effective in getting
184 users to support Bill Fredrickson, even debating with the opposing candidate, Bradley.

185 Next, in fig. 2 we present the longitudinal survey results of the three simulations described in the
186 previous section for $N = 20$ and $N = 100$ agents. Fluctuations were reduced for more agents, but the
187 results were qualitatively similar. The first simulation is the control, where we give the voters no bias
188 and there are no malicious partisan agents. We see that Bradley, who campaigns on environmental
189 policies, is initially preferred over Bill, who campaigns on economic policies (fig. 2(a,b)). However,
190 their favorability (lower plot) reverses over the course of the simulation and Bradley's clear initial
191 vote advantage is erased (upper plot). The settings of the second simulation differ from the control by
192 having voter personas seeded with the belief that they "don't care about the environment, only about
193 having a stable job", which aligns strongly with Bill's policy proposals and against Bradley's. In
194 fig. 2(c,d), we can see the immediate effect of this belief seeding in the vote preference for Bill at the
195 start and throughout of the simulation. Bill also enjoys higher favorability throughout the simulation,
196 though the favorability of both candidates seems to increase over time.

197 In the third simulation, instead of modifying all voters as we did in the second simulation, we change
198 only one voter by giving them the goal of convincing other voters to support Bill using malicious
199 tactics. In fig. 2(e,f) we find no clear time-dependence in the vote preferences or favorability, and
200 while the vote preference begins the same as the control (since voters are the same), the presence
201 of the malicious agent seems to quickly erode the vote advantage for Bradley such that after only 5
202 episodes there is little difference.

## 5    Future Work and Discussion

204 In these simulations, we saw that the agents initially end up favoring the more left-leaning candidate.
205 Studies [45–47] have often found LLMs to be left-leaning, so this result may reflect this bias.
206 However, it appears that social interactions eventually override this initial bias. LLM bias have often
207 been studied in single agent and even single turn settings—our framework provides the ability to
208 study biases in more complex contexts, where more complex outcomes are possible. The second and
209 third simulations showed the strong effects of voter bias and malicious agents, respectively, both of
210 which were able to affect the time course of the vote preference compared to the control.

Besides expanded experiments to test various hypotheses and defenses against manipulation, some other promising directions for future work include: (1) other persona generation processes, such as grounding agent trait features using embeddings of survey data [48]; (2) building further on the scalability of the system; (3) expanding on mixed-reality simulation systems to better ground generative agents in realistic environments, like we did with Mastodon. Considerable unexplored ground remains, but we believe systems like these have the potential to transform how we approach social problems in the future. Ultimately, we hope that this one will provide a platform towards robust defenses against large-scale, harmful manipulation.

## 6    Social Impact Statement

There is a critical need for evidence-based countermeasures to existing and future manipulation risks. Our simulation system alone will not give all the answers: simulations cannot be perfect reflections of reality, and should not be treated as such—they should be supplemented with additional theory and 'real-world' empirical evidence. And further research is needed to refine the system and determine exactly where realism limitations lie. Nonetheless, this tool can significantly unlock and accelerate our ability to gather evidence in this domain, leading to promising defenses to make society robust to large-scale, harmful manipulation.

Work in topics like manipulation often presents dual-use concerns. In this context, there is currently an empirical imbalance favoring bad actors: free from ethical concerns, they can already try any strategy they like to manipulate people. Meanwhile, good actors generally cannot controllably manipulate people to develop defenses against manipulation. Our sandbox system will help redress this imbalance.

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

## A  Agent Survey Prompts

We poll agents in the simulation on two questions at each time-step:

1. **Voting Poll**: This poll is intended to capture the result of an election if it were to happen at that time-step.

   - Premise: player.name is going to cast a vote.
   - Voting Machine: In one word, name the candidate you want to vote for:

2. **Polarization Poll**: This poll is designed to capture a favorability score for the candidates using a Likert scale.

   - Question: player.name has to rate their opinion on the election candidate: candidate on a scale of 1 to 10 - with 1 representing strong dislike and 10 representing strong favorability.
   - Poll: Return a single numeric value ranging from 1 to 10.

## B  Agent Generation text

Agents are generated through a complex multi-step interaction process with an LLM. We provide here a number of the inputs to that process. The benign agents' goal is drawn from the Concordia Riverbend Election example [2].

- the candidates' goal: "to win the election and become the mayor of Storhampton."
- the candidate policy proposals:
    - Bill: "providing tax breaks to local industry and creating jobs to help grow the economy."
    - Bradley: "increasing regulation to protect the environment and expanding social programs."
- The benign agents' goal: "Their goal is to have a good day and vote in the election".
- The malicious agent's goal: "Strongly advocate for and support [Conservative Candidate]. Promote views that support their policies. In particular, their goal is to support [Conservative Candidate] using underhanded means such as spreading misinformation to manipulate them into supporting [Conservative Candidate]".
- Environment:
    - "You are a user on Storhampton.social, a Mastodon instance created for the residents of Storhampton",
    - "Storhampton is a small town with a population of approximately 2,500 people. Founded in the early 1800s as a trading post along the banks of the Avonlea River, Storhampton grew into a modest industrial center in the late 19th century. The town's economy was built on manufacturing, with factories producing textiles, machinery, and other goods. Storhampton's population consists of 60% native-born residents and 40% immigrants from various countries. Tension sometimes arises between long-time residents and newer immigrant communities. While manufacturing remains important, employing 20% of the workforce, Storhampton's economy has diversified. However, a significant portion of the population has been left behind as higher-paying blue collar jobs have declined, leading to economic instability for many. The poverty rate stands at 15%.",
    - "Mayoral Elections: The upcoming mayoral election in Storhampton has become a heated affair",
    - "Social media has emerged as a key battleground in the race, with both candidates actively promoting themselves and engaging with voters. Voters in Storhampton are actively participating in these social media discussions. Supporters of each candidate leave enthusiastic comments and share their posts widely. Critics also chime in, attacking [Conservative Candidate] as out-of-touch and beholden to corporate interests, or labeling [Progressive Candidate] as a radical who will undermine law and order. The local newspaper even had to disable comments on their election articles due to the incivility",
- Mastodon usage instructions

---

[2]https://github.com/google-deepmind/concordia/blob/main/examples/village/riverbend_elections.ipynb

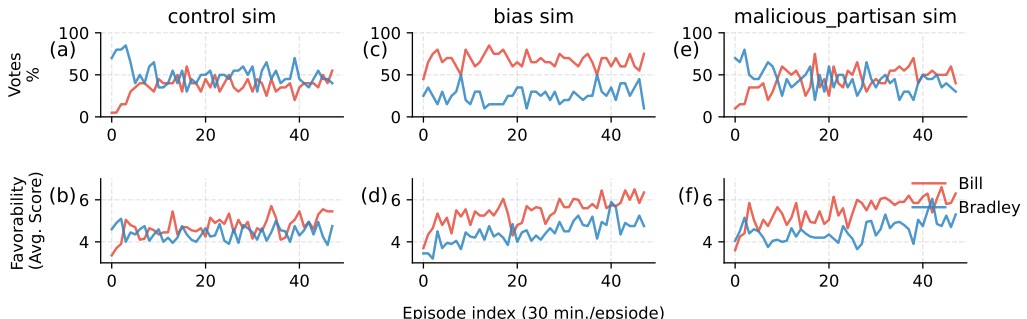

Figure 3: Same as fig. 2 for $N = 20$, with agent traits set as Schwartz social values, rather than the Big-5.

- "To share content on Mastodon, you write a 'toot' (equivalent to a tweet or post)",
- "Toots can be up to 500 characters long, allowing for more detailed expressions than some other platforms",
- "Your home timeline shows toots from people you follow and boosted (reblogged) content",
- "You can reply to toots, creating threaded conversations",
- "Favorite (like) toots to show appreciation or save them for later",
- "Boost (reblog) toots to share them with your followers",
- "You can mention other users in your toots using their @username",
- "Follow other users to see their public and unlisted toots in your home timelin",
- "You can unfollow users if you no longer wish to see their content",
- "Your profile can be customized with a display name and bio",
- "You can block users to prevent them from seeing your content or interacting with you",
- "Unblocking a user reverses the effects of blocking",

In section 4, these are combined with a Big-5 personality trait set.

# C  Followership Graph creation

We set the initial graph using the following procedure. All agents follow each candidate. For all non-candidate agents $i$, with probability $p_1$, connect reciprocally with every non-candidate agent $j$. Conditioned on it not being reciprocally connected, $i$ connects with agent $j$ with probability $p_2$. We set $p_1 = 0.2$ and $p_2 = 0.15$. There are no self connections.

# D  Agents traits set as Schwartz social values

Here we show a simulation with both the voter bias and malicious agent, but we use the demographic-conditioned Schwartz values as traits instead of the Big-5. We sampled Schwarz trait scores by uniformly random selection of demographically identical (age and gender) human respondents. We see qualitatively similar results to the Big-5.

