# OpenReview forum: "Simulation System Towards Solving Societal-Scale Manipulation"
_NeurIPS.cc/2024/Workshop/SafeGenAi — SafeGenAi Poster_

### Official Review · Reviewer_BdfB · 2024-10-08
**Comments on "Simulation System Towards Solving Societal-Scale Manipulation"**

**Rating:** 4
**Confidence:** 4

**Review:**

In this study, the authors simulate multiple LLM agents interacting on social media and participating in a virtual presidential election using the Concordia framework. While this is an intriguing application of Concordia, several aspects of the study require further scrutiny.

Strengths:
The study leverages Concordia in a creative manner to model social interactions and electoral dynamics using LLM agents, which is a novel approach. The simulation of complex, societal-scale interactions using AI agents is an interesting research direction and has the potential to contribute to our understanding of AI’s role in social dynamics and political processes.

Limitations:
1. The paper’s title, “A simulation system towards solving societal-scale manipulation,” suggests the development of an original system, but the paper does not clearly present such a system. If a system component exists, the authors should explicitly articulate what this system is and how it addresses societal manipulation.
2. The paper’s implications for AI safety are unclear, and this is insufficiently discussed in the text. While the simulation is potentially relevant to AI safety concerns, the authors fail to elaborate on how their work contributes to this domain. Furthermore, it is not evident how this study offers insights beyond being a mere application of Concordia. The innovative aspects of this research need to be better defined, and the authors should clarify how it advances LLM or AI research more broadly.
3. One of the most critical limitations of this study is the absence of human benchmarks to compare the behavior of LLM agents against real-world human individuals or collective dynamics. Without this comparison, it is difficult to assess whether the behavior of the LLM agents truly reflects human-like interactions and decision-making processes. Although the authors acknowledge this issue in Section 6, it remains a significant drawback that weakens the study’s validity and limits the potential for real-world applicability.

---

### Official Review · Reviewer_LhTq · 2024-10-09
**Relevant for the workshop. Needs clearer writing.**

**Rating:** 6
**Confidence:** 3

**Review:**

### Summary
This work connects Mastodon, a social media platform, to Concordia, a LLM-based multi-agent framework. By adding clever simulation configurations, the work simulates real-world election environment on three cases: non-biased, inherently biased , and biased with manipulation.

### Pros
- The work addresses a timely question
- The social media framework is well-implemented

### Cons
- Section 4 was unclear-- the authors should put more efforts on the analysis of the results
- Some of the results are surprising--such as manipulation having no real impact on the election results. Analyzing the results in the context of human psychology would be very helpful.
- The simulation assumes that the follow graph can be randomly generated without any prior. However, I feel the follow graph should be grounded on personality traits.

Overall, I believe this is a fit for this workshop.

---

### Official Review · Reviewer_BD1Z · 2024-10-09
**Good to accept**

**Rating:** 7
**Confidence:** 3

**Review:**

The paper presents a simulation system aimed at tackling AI-driven manipulation at a societal scale. It builds upon the Concordia framework, integrating a Mastodon server to simulate social media interactions. The work is significant in its attempt to provide controlled environments to study the risks of disinformation and societal manipulation, which are impractical to test in real-world conditions. Through various means, the authors improve the simulation’s efficiency and scalability, which they demonstrate with election-based case studies involving partisan manipulation.

## Quality

The paper presents a well-executed design and implementation of a complex system. It combines both social media simulation and generative agent-based modeling to mimic societal-scale manipulation. The integration of Mastodon adds a layer of realism to the simulation, which is often missing in related works. The election case study, with different manipulative strategies, demonstrates the system’s potential to uncover manipulation effects, especially through social media platforms.

However, while the methodology is clearly described, certain components could benefit from additional clarity, particularly around how decisions made by agents align with real-world dynamics. For instance, although the use of LLMs to simulate agent behavior is effective, there’s limited discussion on the limitations of the underlying models, like their bias toward particular political stances.

## Clarity

The paper is mostly clear and easy to follow, especially in the way it breaks down its contribution. The figures illustrating the experimental results (e.g., voting preferences and polarization polls) are helpful for understanding the impact of manipulative agents. However, the paper could be clearer in explaining the specific algorithms or metrics used to track manipulation success over time.

## Originality

This paper offers originality in the integration of real social media platforms with generative agent frameworks, which hasn’t been explored at this scale in existing literature. The authors’ use of Mastodon and implementation of analytics tools to measure agent behavior is a novel approach in this space. While there has been previous work on LLM-based agents in simulated environments, this work goes further by addressing the specific challenge of large-scale societal manipulation with concrete tools.

## Significance

The significance of the work lies in its potential application. AI-driven disinformation campaigns are a growing societal problem, and having a framework to test potential defense mechanisms in a controlled, ethical environment is critical. The results from the election case study, demonstrating how manipulative tactics erode political support, underscore the real-world relevance of the research. That said, the paper could have discussed more deeply the practical challenges of deploying such a simulation at even larger scales or across different societal contexts beyond elections.

## Pros:
- Novelty: First to integrate a real social media platform (Mastodon) with an LLM-based multi-agent system.
- Comprehensive Case Study: The election simulation effectively shows how manipulation strategies affect voter behavior.
- Scalability and Efficiency: Significant improvements in runtime and cost, as demonstrated by their use of GPT-4o-mini.
- Open Source: Availability of the codebase fosters transparency and future collaboration in the research community.

## Cons:

- Model Biases: The paper acknowledges some left-leaning biases in LLMs, but doesn’t provide sufficient discussion or mitigation strategies for such issues.
- Limited Real-World Validation: While the simulation framework is promising, more work is needed to demonstrate how findings from simulations can translate into real-world interventions.
- Clarity in Technical Implementation: Certain aspects of the simulation, especially around agent decision-making, could be more thoroughly explained.